# Long-Term Impact of Early-Life Stress on Hippocampal Plasticity: Spotlight on Astrocytes

**DOI:** 10.3390/ijms21144999

**Published:** 2020-07-15

**Authors:** Gürsel Çalışkan, Anke Müller, Anne Albrecht

**Affiliations:** 1Institute of Biology, Otto-von-Guericke-University Magdeburg, 39120 Magdeburg, Germany; guersel.caliskan@ovgu.de; 2Center for Behavioral Brain Sciences, 39106 Magdeburg, Germany; anke.mueller@med.ovgu.de; 3Institute of Pharmacology and Toxicology, Otto-von-Guericke-University Magdeburg, 39120 Magdeburg, Germany; 4Institute of Anatomy, Otto-von-Guericke-University Magdeburg, 39120 Magdeburg, Germany

**Keywords:** astrocyte, early-life stress, maternal separation, juvenile stress, dorsal hippocampus, ventral hippocampus, long-term potentiation, short-term plasticity, gliotransmission, tripartite synapse

## Abstract

Adverse experiences during childhood are among the most prominent risk factors for developing mood and anxiety disorders later in life. Early-life stress interventions have been established as suitable models to study the neurobiological basis of childhood adversity in rodents. Different models such as maternal separation, impaired maternal care and juvenile stress during the postweaning/prepubertal life phase are utilized. Especially within the limbic system, they induce lasting alterations in neuronal circuits, neurotransmitter systems, neuronal architecture and plasticity that are further associated with emotional and cognitive information processing. Recent studies found that astrocytes, a special group of glial cells, have altered functions following early-life stress as well. As part of the tripartite synapse, astrocytes interact with neurons in multiple ways by affecting neurotransmitter uptake and metabolism, by providing gliotransmitters and by providing energy to neurons within local circuits. Thus, astrocytes comprise powerful modulators of neuronal plasticity and are well suited to mediate the long-term effects of early-life stress on neuronal circuits. In this review, we will summarize current findings on altered astrocyte function and hippocampal plasticity following early-life stress. Highlighting studies for astrocyte-related plasticity modulation as well as open questions, we will elucidate the potential of astrocytes as new targets for interventions against stress-induced neuropsychiatric disorders.

## 1. Introduction

Adverse events causing feelings of emotional strain are part of our daily life. Usually, our system adapts well to such stressors. However, when stress endures or is too intense, then adaptation can fail, and stress-induced psychopathologies emerge. However, the levels of stress endurance and intensity that cause a maladaptation are highly individual. Indeed, even after experiencing a potentially life threatening, traumatic event, only a minority of people will develop a long-lasting pathological stress response such as posttraumatic stress disorder (PTSD) [1]. PTSD is characterized by hypervigilance, increased anxiety as well as a pathological, intrusive emotional memory towards the traumatic event. In addition, sleep disturbances, increased aggression and even cognitive problems are observed in PTSD patients; such individuals have a higher risk of developing depressive-like symptoms and substance abuse disorders [2]. One key question in PTSD research is therefore to identify the factors and neurobiological mechanisms that increase the risk for developing PTSD vs. factors that are beneficial and promote resilience. Epidemiological studies have demonstrated that childhood adversity is a prime risk factor for developing PTSD, but also depressive disorders after additional stressors later in life [3]. Indeed, not only prenatally, but also during postnatal life phases, the human brain undergoes particularly sensitive periods for experience-induced plasticity, comprising the pruning of synaptic connections, astrocytic maturation, refinement of cortical neurocircuits via increased inhibition and changes in myelination [4,5,6]. On a behavioral level, these processes result in the maturation of the sensory system in infants, maturation of the motor system enabling a toddler to finally walk and perform complex actions with the hands, and also involve the forming and refinement of higher cognitive functions and control of emotional behaviors during adolescence [5,6,7]. Notably, the maturation of brain areas and circuits associated with emotion control and emotional memory such as the amygdala, hippocampus and medial prefrontal cortex (PFC) undergo restructuring processes during childhood and periadolescence. Importantly, such developmental trajectories are not only observed in humans, but also in the rodent brain during the prepubertal life phase [7], allowing for the use of animal models to study the impact of early life adversity on cognition and emotional behavior later in life. Such animal models either focus on early life stress in the perinatal to preweaning phase by utilizing reduced maternal care, or they model adversity during the late-childhood to periadolescent life phase by introducing juvenile stress (JS) or peripubertal stress to rats and mice. Commonly, such manipulation during earlier life phases affect anxiety- and depression-like behavior later in life, as well as emotional learning and coping with new stressors during adulthood [8,9,10,11,12]. Conceivably, aversive events during these different stages of early life appear to leave permanent traces, potentially leading to (mal)-adaptive modifications in the underlying circuits, which then influence experience-driven plasticity in adult life.

Indeed, in response to new events, neural circuits constantly change in order to adapt their function to new environmental requirements, a process called plasticity, which occurs on different temporal scales. These changes can be expressed as an altered short-term (millisecond-to-minutes) adaptation to a multisensory input, the formation of short-forms of memories or sustenance of working memory (see [13] for an excellent review). Major forms of short-term plasticity (STP) are mediated via transient changes in presynaptic neurotransmitter release or activity-dependent, short-term alterations in the availability or the activity state of postsynaptic receptors. On the other hand, these synaptic alterations can lead to long-lasting (hours to years) and activity-dependent modification of synaptic strength, referred to as long-term plasticity. Substantial evidence indicates that such synaptic alterations are indispensable for the formation of neuronal ensembles and support the formation of long-term memories such as Pavlovian classical conditioning [14,15]. Having used different electrophysiological recordings and stimulation paradigms in vitro and in vivo over the last half century, it is now clear that the same synapse can be modified bidirectionally, leading to either depression or potentiation (LTD or LTP, respectively). Most of the mechanistic insights for the induction, expression and maintenance of such long-term synaptic alterations stem from the hippocampus, due to its fine-layered structure and involvement in episodic memory (see [16] for an extensive review). It is important to note that the induction and expression mechanisms of LTP or LTD can substantially differ, even within different synapses in the hippocampus. For example, while the induction of LTP in the most extensively studied Schaffer collateral (SC)-Cornu Ammonis area 1 (CA1) synapse of the hippocampus depends on the postsynaptic activation N-methyl-D-aspartate (NMDA) receptors (NMDARs), LTP induction in the hippocampal mossy fiber (MF)-CA3 synapse depends on alterations in the presynaptic machinery that mediate neurotransmitter release [16,17]. Lastly, aversive early-life experiences can also alter the capacity of a subsequent neural activity to trigger long-term synaptic plasticity. The term “metaplasticity” is now widely used to refer to this phenomenon; the capacity of synapses to express long-term plasticity depends heavily on the history of their activity [18]. 

To date, most of the studies focused on the neuronal origin of plasticity ignore the nonneuronal factors that may have a substantial impact on the aforementioned temporal scales of synaptic plasticity. Specifically, only a handful of studies have investigated the nonneuronal processes triggered by adverse early-life events that can potentially lead to altered neuronal information processing and synaptic plasticity [19]. At the center of this notion, after post- and pre- synaptic elements, astrocytes have emerged as the third key player in synaptic transmission and plasticity via interacting with synaptic circuits on diverse temporal and spatial scales. As the second major cell type in the brain, long being thought to merely support neuronal metabolism, astrocytes are nowadays acknowledged as fundamental modulators of neuronal signaling [20]; they form the tripartite synapse in conjunction with neuronal pre-and post- synaptic sites,. Astrocytes contribute to the regulation of synaptic function, not only via altering their structure in cooperation with the pre- and post- synaptic components, but also through actively communicating with the neuronal components using distinct signaling mechanisms [21]. 

In this review, we first discuss the multiple processes by which astrocytes can control short- and long-term plasticity at the tripartite synapse. Indeed, diverse changes in STP and LTP have been observed, especially in subregions of the hippocampus following early life adversity. We provide a short overview of the plasticity-related changes evolving in adulthood after a history of reduced maternal care or JS, and discuss recent studies highlighting the contribution of astrocytes to (mal)-adaptive plasticity. Thus, we can identify current gaps in knowledge and themes for further research regarding the putative mechanisms of a lasting astrocytic modulation of plasticity after early-life stress.

## 2. The Numerous Ways in which Astrocytes Modify Plasticity

While astrocytes comprise the second largest cell population in the central nervous system and are the predominant glial cell type besides oligodendrocytes and microglia, they have only received a modest degree of attention in studies of neurobiological processes in health and disease. In gray matter, protoplastic astrocytes are widely distributed and form highly ramified processes which permeate the extracellular space between neuronal cells and contact neuronal cell bodies, blood vessels and synapses, in order to metabolically support neurons, regulate brain homeostasis and also modulate neuronal signaling by releasing neuromodulatory factors or regulating neurotransmitter concentrations at synaptic sites (see also [20,22,23,24] for general review). Indeed, more and more studies have described the profound modulative capacities of astrocytes to shape neuronal signaling, both on a syncytial and on a single synapse level. By forming the tripartite synapse, astrocytes are an integral partner for the neuronal pre-and post- synaptic site [25]. Synapses with fine perisynaptic astrocytic processes (PAP) contacting pre- or post- synaptic sites can be found in many brain regions, although to different extents. About 60% of CA1 rodent hippocampal synapses were found to have PAP contact, although contact formation by PAP seems to be a dynamic process [26,27]. Indeed, PAPs can be highly motile and the number of synapses with PAP contacts appears to correlate with synapse complexity and strength [27,28]. Although astrocytes contribute to synapse development as well, the formation of PAPs, in contrast, follows synaptic formation by neurons. In rats, astrocytes mature at around postnatal day (PND) 21–28 and emerge as highly ramified cells with clearly distinguishable territories that do not overlap with their neighboring astrocytes [4,29].

In contrast to the clearly structured neuronal pre- and post- synaptic sites, astrocytic PAPs are rather undefined structures and, in comparison to neurons, mostly unresolved on a molecular level due to the difficulty in accessing these finely structured processes. Whereas astrocytes are electrically nonexcitable cells, several other components of PAPs mark them as players in neurosignaling. Astrocytes can sense the release of several neuromodulators, as well as neurotransmitters, by expressing a high variety of receptors. Receptor subtypes comprise metabotropic glutamate receptors that were also identified on astrocytic PAPs [30], as well as GABA receptors, purinergic receptors and β2-noradrenergic receptors, among others [24,31,32,33]. These receptors are activated by changes in neuronal activity and enable the modulation of astrocytic intracellular calcium levels to occur, which triggers downstream signaling cascades. Subsequently, astrocytes are able to signal back to neurons using gliotransmitters or react by structural adaptations and can thereby modulate neuronal activity (see [20,34] for a general review). In the following sections, we will briefly introduce some of the manifold ways in which astrocytes can influence neuronal activity and synaptic plasticity.

### 2.1. Structural Adaptations of PAPs

One of the activity-dependent adjustments is structural adaptations of PAPs. While the induction of LTP in the hippocampus enhances dendritic spine volume and postsynaptic density size in neurons, the number of synapses contacted by astrocytic elements and the size of the astrocytic contact area at pre-and post- synaptic sites is increased as well [35,36]. These structural adjustments of astrocytic PAPs are likely mediated by glutamate-induced calcium signaling in hippocampal astrocytes [35,37,38,39]. Although some studies were not able to prove directed enhancement of PAP motility with increased neuronal activity in hippocampal slices or at corticostriatal synapses [28,40], further examples from tissues such as the barrel cortex [41], amygdala [42] and hypothalamus [43] were able to report structural adaptations of PAPs following changes in neuronal activity. Although tissue- or synapse-specific consequences might be heterogeneous, a general mechanism of activity-dependent PAP adjustment appears probable. PAPs support neurotransmitter uptake by expressing specific transporters, but they also release gliotransmitters; therefore, structural rearrangements of PAPs have the potential to alter synaptic plasticity on multiple levels. 

### 2.2. Neurotransmitter Uptake and Metabolism

The localization of both glutamate and GABA transporters in PAPs allows the uptake of transmitters into astrocytes to occur, and thereby terminates neuronal signaling at synaptic and extrasynaptic receptors [44,45,46]. Since excess glutamate is neurotoxic, the uptake of glutamate via astrocytic transporters GLAST (EAAT1) and GLT-1 (EAAT2) is important for maintaining synaptic function. Indeed, several studies have reported that astrocytes regulate their glutamate transporter expression and localization as an adaptive mechanism to neuronal activity changes. The expression of GLT-1, for example, is enhanced in organotypic spinal cord cultures, as well as in the barrel cortex, upon increased neuronal activity. Furthermore, in developing astrocytes, neuronal activity leads to the clustering of GLT-1 close to synaptic sites [41,47,48]. Thus, increased availability and activity of GLT-1 may allow increased glutamate uptake to occur in the vicinity of synaptic sites, which feeds back onto neuronal signaling. Accordingly, the downregulation of astrocytic GLAST1 in the hippocampus by the proinflammatory protein IFN-beta promotes LTP induction and memory formation, probably by increasing the glutamate concentration at the synaptic cleft [49]. In parallel, glutamate at the tripartite synapse also activates metabotropic glutamate receptors in astrocytes upon neuronal stimulation, thereby triggering intracellular signaling cascades that adjust the rate of glutamate uptake [50,51]. Via activation of metabotropic glutamate receptors, both astrocytes and neurons regulate the release of glutamate in a feedback loop [52,53,54]. This may explain why the impaired activation of GLT-1 and GLAST by amyloid-beta oligomers also results in decreased hippocampal LTP [55]. For the uptake of GABA, astrocyte-specific uptake transporters exist as well. Using pharmacological blockers, it has been demonstrated that this transporter, GAT-3, determines ambient GABA levels in the neocortex [56]. In the hippocampus, GAT-3 is expressed extrasynaptically, and thus, is only effective under high GABA concentrations, likewise preventing the detrimental effects of neurotransmitter spill-over [57]. GAT-3 is furthermore colocalized with GLT-1, especially within the CA1 and CA3 area of the hippocampus, and under high neuronal activity, both transporters may interact to shape network activity [58]. Accordingly, reducing astrocytic glutamate release by expressing tetanus neurotoxin specifically in astrocytes leads to the suppression of oscillatory network activity, both in vitro and in vivo [59]. After uptake of glutamate into astrocytes, the enzyme glutamine synthetase metabolizes glutamate to glutamine, which is then shuttled back to neurons to synthesize new glutamate. Within inhibitory neurons, glutamate is decarboxylated by the enzymes GAD65 and GAD67 to the inhibitory neurotransmitter GABA [60]. Since glutamine stores are low in close proximity to GABAergic terminals, glutamine synthetase expression can limit GABA release in the hippocampus, especially under heightened neuronal activity [61]. In this line, the coordinated activation of astrocytes via astrocytic GABA-B receptors and of GABAergic interneurons is required in vivo for the maintenance of oscillatory hippocampal activity at the low-gamma (30–50 Hz) and theta range [62]. 

### 2.3. Gliotransmitter Release

Glutamate is not only taken up by astrocytes; it is also among the substances released by these cells in response to enhanced neuronal activity, the so-called gliotransmitters [63]. Glutamate can be released from astrocytes via the reversal of glutamate transport or via hemichannels and ion channels. In addition, elevation of cytosolic Ca^2+^ from internal stores induces glutamate release by regulated exocytosis [64,65,66]. Astrocytic glutamate, in return, has been shown to activate neighboring neurons in single neuronal cultures, to influence neuronal activity and to enhance synaptic strength at excitatory synapses in hippocampal slices [65,67,68]. Astrocytic glutamate release further contributes to NMDA-receptor-dependent LTD formation and to setting the threshold for LTP-induction in the hippocampus [69]. Beside glutamate, other gliotransmitters such as ATP also affect neuronal signaling. The release of ATP from astrocytes appears to profoundly modulate memory-associated gamma-range (30–80 Hz) neuronal network oscillations. Specifically, the optogenetic-activation of astrocytes results in the release of ATP that alters the excitability of CCK+ interneurons and pyramidal neurons, which, in turn, reduces kainate-induced gamma oscillations in vitro [70]. 

Controversial, however, is the role of astrocytes in providing D-Serine, a cofactor of NMDA receptors that supports the establishment of LTP at the mature glutamatergic synapse. It was long thought that astrocytes were the sole source of D-Serine synthesis and release upon increase in astrocytic Ca^2+^ transients [71]. While several studies have reported D-Serine in astrocytic vesicles [72], neurons are the major source of D-Serine synthetase, a serine racemase that catalyzes the conversion of L-Serine to D-Serine [73,74]. This may point towards a neuronal mechanism of cofactor release and subsequent NMDA receptor activation. Nevertheless, concurrent roles of the storage and release of D-Serine from astrocytic terminals seem possible [75,76,77,78]. However, astrocytes provide the substrate for serine racemase, L-Serine, thereby sustaining neuronal D-Serine synthesis and indirectly affecting synaptic potentiation. The removal of L-Serine or inhibition of 3-phosphoglycerate dehydrogenase (Phgdh), the astrocytic key enzyme for L-Serine and glycine synthesis, also depletes D-Serine levels, and subsequently leads to reduced NMDA receptor potentials in hippocampal slices [79]. Beside the molecules that are categorized as gliotransmitters, other astrocytic proteins and signaling molecules can also modulate neuronal activity and influence long-term potentiation, including the matricellular protein SPARC [80] and the water channel aquaporin-4 [81].

### 2.4. Metabolic Support

Another metabolite that affects long-term memory and LTP formation is lactate, that sustains neuronal energy demands during neuronal activity. Astrocytes store metabolites for the TCA cycle in the form of glycogen, thereby providing a local source for the energy used by synapses that can be released upon enhanced neuronal activity [82]. In the current model of neuron–astrocyte metabolic coupling, enhanced neuronal activity leads to elevated potassium levels and a downstream cascade that subsequently induces increases in cAMP levels and the activation of astrocytic glycogenolysis [83]. Lactate, as the product of glycogenolysis, is then transported via a set of monocarboxylate transporters to neurons for energy production in neurons [84]. Increasing lactate levels further enhance the NADH/NAD+ (NAD: Nicotinamide adenine dinucleotide) ratio in neurons, leading to a subsequent increase in Camk2 (Ca^2+^/calmodulin-dependent protein kinase2) phosphorylation and support of LTP induction [85]. Such a molecular process may provide the basis for the observed NMDA receptor potentiation and the induction of plasticity-related gene expression [86]. Accordingly, the disruption of glycogenolysis or the transport of lactate results in an impairment of long-term memory and LTP, further highlighting the role of the complex metabolic coupling between neurons and astrocytes [84,87].

### 2.5. Connexin Hemichannels

One functionally important feature of astrocytes is their connection with each other via hemichannels formed by a hexameric ring of connexin proteins (Cxs). When hemichannels between neighboring astrocytes face each other, they form “gap junctions”, supporting direct intercellular communication via the exchange of ions and small molecules within the astrocytic network, thereby enabling crosstalk with neuronal networks over a wide range [88]. The main forms of Cxs expressed in astrocytes are Cx30 and Cx43. They play a crucial role in the buffering of ions, propagation of calcium waves and homeostatic regulation of synaptic transmission and neuronal excitability [89,90]. Interestingly, neuronal activity can shape the expansion of an astrocytic network formed by Cx30 and Cx43, and thereby enhance astrocytic coupling and glucose distribution [91]. On the other hand, nonjunctional forms of hemichannels also exist and mediate the release and uptake of diverse ions and metabolites, such as gliotransmitters (ATP, glutamate etc.) and D-serine to and from extracellular space [89,92]. Mice deficient in Cx30 demonstrated that these factors regulate astrocytic PAP invasion, independent of its hemichannel function, thereby regulating glutamatergic neurotransmission at the tripartite synapse [93].

Considering the tight interaction of neurons and astrocytes in the establishment and induction of synaptic plasticity, further studies are needed to understand the role of astrocytes in the long-term adjustment of synapse strength, and might help to unravel the astrocytic role in memory storage. In this way, new therapeutic targets for the treatment of stress-induced memory impairments and associated neuropsychiatric disorders such as anxiety and depression could be identified. 

## 3. Plasticity in Early Life Stress Models: Diversity across the Dorsoventral Axis of the Hippocampus

Animal models are indispensable tools to investigate the neurobiological traces of early-life stress in the adult brain. Several procedures have been developed in rodents to mimic an adverse childhood. One of the most prominent models concentrating on the preweaning life phase is maternal separation (MS). Typically, rats and mice are weaned off their mothers at the age of PND 21; the repeated separation of pups from their mothers during their first two postnatal weeks can evoke increased anxiety- and depression-like behavior later in life. However, outcomes vary greatly depending on the exact age of the separated pups, the duration of the separation sessions and whether pups we kept alone or grouped during separation sessions [8,9]. Moreover, separation also influences the behavior of the mothers, and may induce increased maternal care once the pups return to the nest after their separation session [94], thereby inducing beneficial cognitive and emotional behaviors and increased stress resilience in their offspring later in life. Other protocols aim to induce fragmented maternal care, for example by providing only limited bedding and nesting material (LBN model) in a more stable manner. Many variants of the LBN model exist across laboratories (see [12] for review) which frequently elicit increased anxiety, depression-like behavior and increased fear learning in LBN-exposed rats and mice later in life [12]. Similar behavioral sequelae of adverse early life events are also observed when stressors are delivered later in childhood, after weaning at PND 21 and until about the end of puberty at around PND 42. Here, many JS paradigms using relatively short duration manipulations (e.g., three different stressors on PND 27-29) induce increases in anxiety-like behavior, increased fear memory with deficits in fear extinction as well as disturbed stress coping in avoidance tasks [11]. More enduring stress exposure during the postweaning/prepubertal life phase may instead cause depressive-like behavioral traits and affect social interaction later in life [10,95].

Following all these models of early-life stress, long-term changes in plasticity were observed and are particularly well studied in the hippocampal formation. It is now well-established that the dorsal and ventral portions of the hippocampus show differential anatomical connectivity and support diverging functions [96]. While the dorsal hippocampus (DH) supports spatial memory and higher cognitive functions, the ventral hippocampus (VH) is mainly involved in anxiety, stress response and emotional memory formation [96,97,98,99]. Since early-life adversity is associated with lastingly enhanced anxiety and stress-induced memory deficits, we will attempt to mention whether the plasticity measurements were obtained from the VH or DH, whenever possible. Accumulating evidence suggests that MS-induced early-life stress using different rodent models results in impaired LTP, specifically at the SC-CA1 synapse of the hippocampus. MS for 3 h daily between PND 2 and PND 14 reduces high frequency stimulation (HFS) -induced LTP at the dorsal SC-CA1 synapses without any alteration in baseline synaptic transmission in adulthood (PND > 60) [100]. In line with the notion that DH is involved in spatial memory, rats that went through MS show a deficit in spatial memory in the Morris Water Maze (MWM) task [100]. Another study using a longer MS protocol (PND 1-20) demonstrated a similar reduction in the dorsal SC-CA1 LTP in vivo and a spatial deficit in the MWM task in adulthood [101]. Accordingly, the LBN model of early life stress also leads to an LTP impairment at the dorsal SC-CA1 synapse during adulthood [102,103]. Impaired LTP after LBN is associated with a spatial deficit in the MWM task and a reduced expression of GluN2B-containing NMDA-Rs that are crucial for the induction of LTP in this region [102,103]. A deficit in LTP at the SC-CA1 synapse of DH after LBN is also evident in adolescence and is accompanied by an enhanced LTD [104]. Similarly, the decrease in the HFS-LTP at the SC-CA1 synapse after MS (PND 1–14, 3 h daily) is present in adolescence (PND 28–35) [105] and continues until old age (70 weeks) [106]. However, a milder MS protocol (PND 1–7, 1 h daily) is not able to reduce HFS-LTP [107], but leads to the enhancement and delay in developmental decay of LTD induced by low frequency stimulation (LFS) during adolescence [108]. Remarkably, reduced LTP and enhanced LTD appear to be present, even in the offspring of male mice that were exposed to MS [109]. These data suggest that MS and/or LBN might affect the mechanisms that are required for the maturation of bidirectional plasticity at the SC-CA1 synapse during adolescence, and lead to an intergenerational transmission of abnormal plasticity. Furthermore, the severity and duration of the MS appear to be crucial factors that determine the presence of an LTP deficit at the dorsal SC-CA1 synapse later during adulthood. Interestingly, even relatively longer periods of MS (PND 2–20, 4 h daily) do not affect theta-burst stimulation (TBS) -induced LTP at the ventral SC-CA1 [110]. Similarly, no effects were observed on baseline transmission and STP, evident by comparable input–output curves and paired pulse responses, respectively. Together, these results suggest that MS predominantly affects LTP in the SC-CA1 synapse of the DH, in comparison to the ventral counterpart. Interestingly, the same study found a strong reduction in STP and LTP at the ventral MF-CA3 synapse, indicating that the effects of MS on the VH physiology are mainly mediated via CA3 subregion [110].

Another hippocampal region that functions as the gatekeeper of the hippocampus is the dentate gyrus (DG) [111]. Several earlier studies using in vivo recordings from freely moving rats provided evidence for an enhanced HFS-LTP upon stimulation of perforant path (PP) fibers originating in the entorhinal cortex in both adolescence (PND 30) and adulthood (PND 70–90) after MS (PND2–9, 1 h daily) in a sex-dependent manner [112,113,114]. In contrast, both early (PND 2-9) and late (PND 14–21) MS (6 h daily) impair HFS-induced LTP in the DG in anesthetized rats in vivo [115]. This discrepancy between these studies might be related to the differences in the brain states during induction of the LTP in freely behaving vs. anesthetized rats, and the duration (1h vs. 6h) of MS during each day. Finally, history of MS strongly interferes with acute stress-induced LTP in the PP-DG pathway during both adolescence and adulthood in freely moving rodents [116,117]. Together, it is evident that most studies have been focused on the impact of preweaning stress on DH plasticity, ignoring the ventral counterpart. Considering the substantial involvement of VH in affective behavior, upcoming studies should aim to reduce this knowledge gap.

In contrast to the studies investigating the impact of MS on hippocampal physiology, the differential impact of stress during the postweaning/adolescence period on VH vs. DH plasticity has been investigated more extensively. We and others have consistently shown that stress exposure (PND 27–29 or PND 28–30) during juvenility prompts the induction and maintenance of LTP at the ventral SC-CA1 synapse, while decreasing it somewhat in the dorsal counterpart later in adulthood (PND 60–120) [118,119,120]. Strikingly, the combination of JS with acute stress during adulthood further promotes the stabilization of this discrepancy in the ability to express LTP in the VH. vs. DH [118]. A similar disparity is also observed for the magnitude of LTD after adult stress when combined with previous stress during juvenility, expressed as an enhanced LTD in the DH and a lack of LTD in the VH [118]. Reduced dorsal LTP is even observed in old animals (15 months) upon chronic social stress during adolescence [121]. On the other hand, JS appears to have minimal effect on LTP/LTD at the dorsal DG. However, a strong reduction in STP is evident by reduced paired pulse responses measured during adulthood [122]. Thus, we aimed to test whether JS would have a differential impact on synaptic transmission and plasticity in the ventral DG (Figure 1; see Appendix A for details). We obtained our measurements from medial PP to DG synapse using horizontal brain slices including transverse-like sections of the VH. To our surprise, we did not observe any alteration in baseline synaptic transmission or STP in adulthood (PND 84), evident by the absence of any effect of JS on input–output curves and paired-pulse responses, respectively. Similarly, JS did not alter HFS-induced LTP measured at adult age.

Taken together, stress during this sensitive period affects bidirectional plasticity in the “cognitive” DH and “affective” VH in a contrasting fashion. Such an opposing impact may induce diverging synaptic scaling/homeostatic plasticity in brain regions with differential functional and anatomical connectivity to VH or DH [11,123].

## 4. Astrocytic Alterations after Early Life Stress

As stated above, astrocytes can modulate plasticity at the tripartite synapse in multiple ways. The majority of studies investigating the impact of early life stress on astrocytes have focused on the expression of astrocyte-specific markers, such as glial fibrillary acid protein (GFAP) and the calcium binding protein S100ß in brain areas involved in emotional behavior control, while only a handful have assessed specific astrocytic functions (Table 1).

Within the DH of adult rats, preexposure to MS reduced GFAP expression [124,125] or the number of GFAP(+) cells [126] in most studies (but see also [127]). A reduction of GFAP immunoreactivity was also observed after fractionated maternal care with the LNB protocol [128]. In rats with a history of stress during the postweaning/peripubertal phase, the number of GFAP(+) cells was reduced in the PFC [130], and GFAP mRNA levels were reduced in the dorsal DG after 3 days of variable JS [122]. In adulthood, prolonged stress exposure only reduces GFAP immunoreactivity in a similar manner in the basolateral amygdala [131] and in the DH [134]. While reduced GFAP immunoreactivity as a consequence of stress may be related to a reduced number of astrocytes, a parallel analysis of an astrocyte-specific marker S100ß after chronic stress in adulthood demonstrated an increase [134]. However, baseline expression differences exist between GFAP and S100ß, with S100ß being not exclusively expressed in astrocytes [136]. Recent studies providing additional structural analyses have suggested that increased chronic, mild, unpredictable stress (CMUS) led to a reduced structural complexity of astrocytes in the dorsal DG [135]. Acute swim stress in adulthood further resulted in astrocytic hypertrophy and an increase of calcium levels in astrocytic microdomains [90]. The same study also demonstrated reduced expression of the astrocytic gap junction channel proteins connexin 30 and connexin 43, which were associated with impaired LTP. Interestingly, prolonged stress exposure during adulthood rather increased connexin 43 in the DH [132].

Together, instead of general markers of astrocytes, the analysis of specific astrocytic proteins may hold greater potential to reveal functions of astrocytes as sequelae of stress. While such studies are still ongoing for models of preweaning early life stress, we started to investigate the key players of astrocyte-neuron interactions after variable 3-day JS. We found that preexposure to JS reduced the expression of GLT-1 and GAT-3 in the granule cell layer of the dorsal DG in adulthood. These changes were associated with an increased paired pulse inhibition, a form of STP. Increased PPI was pharmacologically mimicked by blocking of GAT-3 in stress-naive rats. Thus, the reduced uptake of GABA into astrocytes appears to increase inhibition in local dorsal DG circuits as a long-lasting consequence of JS [122]. Interestingly, GLT-1 within the dorsal DG is also reduced after CMUS in adulthood [133], suggesting that similar astrocytic mechanisms could contribute to reduced plasticity after chronic stress in adulthood. With a similar stress exposure protocol, we also investigated dorsal and ventral SC-CA1-LTP, and found that the previously reported increase of LTP in the ventral CA1 of rats with a history of JS can be mimicked with a pharmacological blockage of the enzyme glutamine synthetase, which breaks down glutamate to glutamine within astrocytes. Increased LTP could be normalized in slices of juvenile stressed rats by supplementing glutamine. The expression of glutamine synthetase mRNA was also reduced in the *stratum radiatum*, the layer of SC synapses in CA1 [120]. Both studies demonstrated that a combined analysis of functional astrocytic domains may help to understand how this cell type shapes lastingly plasticity. Moreover, they indicated that different processes may take place in parallel in different ventral and dorsal hippocampal subareas (Figure 1). 

## 5. Conclusions and Future Directions 

Stress early in life is one of the most important risk factors for developing psychopathologies in adulthood, like depression and anxiety disorders. Studies in humans and rodents have demonstrated that early-life stress shapes how individuals respond to stress later in life [1,137,138]. Using short- and long-term models of cellular plasticity, it became evident that early-life stress shapes plasticity in adulthood; one of the core questions is therefore which neurobiological correlates exist for such lasting impacts on plasticity. Assessing astrocytes in this context comprises a research area with a high potential to discover new targets for the therapy of stress-induced neuropsychopathologies.

Indeed, as summarized above and by others, astrocytes are powerful modulators of neuronal plasticity, but neuronal activity itself shapes the interaction of neurons and astrocytes at the tripartite synapse as well [20,22,23]. PAPs provide a structural correlate of astrocyte–neuron interactions and allow astrocytes to take up and metabolize neurotransmitters at and around the synapse. Moreover, by expressing their own metabotropic neurotransmitter receptors, astrocytes can sense neuronal activity, adapt their own activity by increasing intracellular calcium levels and actively modulate synaptic transmission, for example by the release of gliotransmitters. In addition, astrocytes are connected among each other via gap junctions, and therefore provide the basis for adapting neuronal activity across a wide population of synapses and for modulating network activity as an important feature of metaplasticity, especially in the hippocampus and PFC.

To date, most of the studies investigating the lasting consequences of stress across different life phases on astrocytes have merely assessed the expression of astrocytic markers such as GFAP, finding, overall, a reduction (see Table 1). One major question is now whether such a change in astrocytic markers is epiphenomenal, or whether it has functional implications and contributes to the dysregulation of synaptic activity. To that end, studies are required that methodologically cross the line from descriptive to functional investigations of astrocytes after early life stress. Such studies would require a methodical investigation of different astrocyte functions like neurotransmitter uptake, gliotransmission and metabolic support within different early life stress models, and their contribution to hippocampal plasticity. To date, we are only at the beginning of such a research agenda. Using JS as a model, we showed that the observed reduction in GAT-3 in the dorsal dentate gyrus indeed shifts inhibition in STP protocols after JS, but not stress in adulthood [122]. In parallel, JS appears to affect long-term plasticity in the ventral CA1 by affecting glutamine synthetase and the glutamate/glutamine cycle [120]. Similar studies that combine expression analyses of astrocyte-specific factors with pharmacological interventions are, unfortunately, yet lacking for other early-life stress models such as MS or limited nesting.

Given the functional complexity of astrocyte–neuron interactions, an altered neurotransmitter uptake and metabolism as a consequence of early life stress may not be the only astrocytic functional domain contributing to stress-induced changes in LTP and STP. For example, metabolic coupling and lactate shuttling might be affected, or the release of other gliotransmitters such as ATP or D-serine could be affected as well. Another as yet unresolved question is whether early-life stress has stable long-term effects on astrocyte structure and interactions via gap junctions. First insights from adult stress models underline the modulative capacity of stress on astrocytes. Acute adult stress appeared to induce an astrocytic hypertrophy and an increase in astrocytic microdomain calcium levels, while the expression of connexin proteins was reduced [90]. Other studies using chronic stress exposures in adults showed a reduced structural complexity of astrocytes [135]. Thus, structural assessments of astrocytes over time following early life stress events such as MS, limited nesting and bedding or postweaning JS may widen our view on the lasting impact of stress on astrocytes.

Moreover, instead of assessing isolated astrocytic factors that could be potentially altered following stress, transgenic animal models are available that allow cell-type-specific metabolic tagging of newly synthesized proteins to be performed that discriminate between neuronal and glial proteomes, and thereby make possible the identification of global astrocyte specific changes following stress [139,140]. Using the RiboTag mouse, Murphy-Royal et al. were able to identify altered expression of astrocytic genes following stress exposure in adult mice [90]. Similar approaches can be utilized after early-life stress, and might help to identify further gene expression changes underlying plasticity mechanisms that add to the astrocytic modulation of synaptic strength in hippocampal LTP or LTD. In addition, cell-type-specific knock-down of candidate proteins or the chemogenetic silencing of cell populations [141] provide excellent tools to study the contribution of astrocytes to STP and LTP upon early-life stress exposure. In this line, the site of intervention is an important variable in future research. 

As the findings from the JS studies demonstrate, there is a high regional specificity in the impact of stress on hippocampal plasticity (see also Figure 1). Increased plasticity in the VH has been specifically observed after stress (e.g., [118]) in structures that are anatomically and functionally connected to emotional-relevant areas such as the mPFC and the amygdala [96]. It is tempting to speculate that a stress-induced “dynamic routing” of information processing to the VH may be related to the increased anxiety, altered fear memory and increased depression-like symptoms observed after early-life stress (e.g., [11]). Thus, next to studying STP and LTP after early life stress and the associated role of astrocytes, further studies are needed that also investigate behavioral alterations after stress. Again, pharmacological and genetic approaches such as cell-type-specific knock down and chemogenetic silencing might be of tremendous help in investigations of the contribution of astrocytic factors to phenotypes induced by early-life stress.

In summary, identifying the relevant molecular mechanisms targeted by early-life stress in astrocytes would make it possible to identify possible pharmacological targets that attenuate stress-induced alterations in astrocytes; thereby, new therapeutic avenues for the treatment of stress-induced psychopathologies such as anxiety disorders and depression could be established and tested in translational settings.

## Figures and Tables

**Figure 1 ijms-21-04999-f001:**
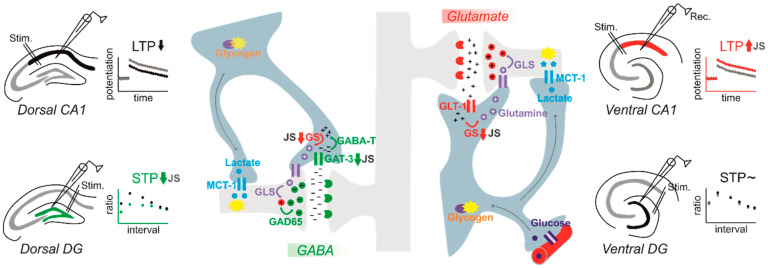
Juvenile stress lastingly affects neuron–astrocyte interactions, which are linked to region-specific changes in short- and long-term plasticity in the hippocampus. Exposure to variable stress during juvenility (JS) lastingly reduces the expression of the astrocyte-specific gamma-aminobutyric acid (GABA) transporter GAT-3, leading to increased inhibition/reduced facilitation in short-term plasticity (STP) protocols (see left side, green). This effect was specifically observed in the dorsal dentate gyrus (DG; see left side, green), while no changes were observed in the ventral DG (see right side). Indeed, reduced astrocytic uptake of GABA would increase inhibition at neuronal synapses. Within astrocytes, GABA is further converted to glutamate via the enzyme GABA transferase (GABA-T). Glutamate is then converted to glutamine via glutamine synthetase (GS) and shuttled back to neurons, where new glutamate is synthetized by the enzyme glutaminase (GLS). In inhibitory neurons, Glutamate decarboxylase GAD65 (and the isoform GAD67) converts glutamate to GABA. The glutamate/glutamine cycle is also active in astrocytes around glutamatergic synapses (right side, red), where glutamate is taken up into astrocytes by the transporter GLT-1. Interestingly, within the ventral CA1 subregion of the hippocampus, expression of GS was reduced, leading to increased LTP specifically in this region (right side, in red), while dorsal CA1 LTP is reduced by JS (left side, black; the association with astrocyte-neuron-interaction is unclear). Next to neurotransmitter uptake, astrocytes interact with neurons in multiple ways. For example, in contrast to neurons, astrocytes can store glucose by transforming it to glycogen and provide neurons with lactate as a source of energy. Lactate is shuttled to neurons via specific transporters, e.g., monocarboxylate transporter 1 (MCT1). While the blocking of MCT-1 has been shown to affect LTP and memory, the impact of JS on this system is still unclear and remains to be investigated in future.

**Table 1 ijms-21-04999-t001:** Impact of stress across different life phases on astrocytic factors.

	Protocol	Testing	Effect	Reference
	MS 3h/d PND 2-15	Around PND 70	⇩ GFAP mRNA in PFC	[124]
	MS 4h/d PND 1-21	PND 100	⇩ GFAPir in PFC, ACC, Striatum and dorsal Hippocampus	[125]
	MS 4h/d PND 1-14	3-5 month	⇩ GFAP(+) cells in PFC and dorsal Hippocampus	[126]
	MS 3h/d PND 1-10	PND 60	⇧ GFAPir in Hippocampus	[127]
	LBN PND 2-9	10 months	⇩ GFAPir dorsal Hippocampus	[128]
	MS 6h/d PND 15-22	12 weeks	⇧ GFAPir in Locus coeruleus (females)	[129]
**Postweaning/peripubertal stress**	Noise exposure PND 21-35	PND 90	⇩ GFAP(+) cells in PFC	[130]
	Variable JS PND 27-29	PND 74	⇩ GFAP/⇩ GLT-1/⇩ GAT-3 mRNA in dDG granule cell layer	[122]
	Variable JS PND 27-29	PND 74	⇩ Glutamine synthetase mRNA in vCA1	[120]
**Adult stress**	Acute swim stress	adult	⇧ astrocyte hypertrophy ⇧ microdomain calcium ⇩ Connexin-30 and -43 in somatosensory cortex	[90]
	10d immobilization 2h/d	adult	⇩ GFAP ir in BLA but not dCA3(+) cells in PFC	[131]
	10d immobilization 2h/d	adult	⇧ Connexin-43 in hippocampus	[132]
	18 d CMUS	adult	⇩ GLT-1 mRNA in dDG	[133]
	21 d CMUS	adult	⇩ GFAP ir ⇧ S100ßir in dorsal hippocampus	[134]
	CMUS	adult	⇧ S100ßir in dorsal DG, but reduced structural complexity	[135]

MS: Maternal separation; LBN: limited bedding and nesting; CMUS: Chronic mild unpredictable stress; JS: juvenile stress; PND: postnatal day.

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
