# Peer review of "Long-Term Impact of Early-Life Stress on Hippocampal Plasticity: Spotlight on Astrocytes"

_ijms, 2020, doi:10.3390/ijms21144999_

Round 1

Reviewer 1 Report

In this manuscript the authors describe current findings on how astrocytes can influence neural activity and synaptic plasticity in the hippocampus and how early-life stress affects the hippocampal synaptic plasticity in several animal model. Furthermore, the authors summarize alterations of astroctyes induced by early-life stress to emphasize a contribution of astrocytes to change of synaptic plasticity in the hippocampus after stress. The present work provides knowledge of astrocytic modulation in stress-induced changes of neural plasticity and may be used to get insight on the pathogenesis of anxiety or depressive disorders.

Although it has potential to be considered for publication, several issues should be addressed.

  1. The authors should specify the reference for a statement: from page 2, line 47, beginning with the words "refinement ~" to page 2, line 48, ending with the words " ~ in myelination".
  2. In page 4, line 164, is "adaptive mechanisms of [46]" a typo?
  3. From page 4, line 164 to line 167, it is obscure why this research data can be an example of astrocytic function to modulate neuronal activity in adaptive mechanism. The data only describes the expression of GLT-1 in some neural tissues and conversely demonstrates that neural activity changes GLT-1 expression pattern in astrocytes. 
  4. The contents of conclusions and future directions are ambiguous. The authors seem to emphasize an importance of functional study on astrocyte in juvenile stress model. However, the point is not clear especially from page 10, line 439 to the end.
  5. Overall, the neural plasticity of hippocampus in stress animal model is a main content of this manuscript although they highlight astrocyte. I think the authors had better add the "hippocampus" in the title.

Author Response

  1. The authors should specify the reference for a statement: from page 2, line 47, beginning with the words "refinement ~" to page 2, line 48, ending with the words " ~ in myelination".

We thank the reviewer for her/ his valuable comments. Indeed, the reference number 4 appears somewhat misplaced in midsentence. In this review, Akdemir et al. describe developmental perspectives of astrocytogenesis. However, in this sentence more events that characterize sensitive periods of plasticity during development are described, including synaptic pruning, altered excitation/ inhibition balance and myelinization.  We added the respective references to the end of the sentence (line 49).

  1. In page 4, line 164, is "adaptive mechanisms of [46]" a typo?

Thank you for the notification. We changed the typo and inserted: “..to neuronal activity changes.” to clarify the statement on page 4, line 165-166,

  1. From page 4, line 164 to line 167, it is obscure why this research data can be an example of astrocytic function to modulate neuronal activity in adaptive mechanism. The data only describes the expression of GLT-1 in some neural tissues and conversely demonstrates that neural activity changes GLT-1 expression pattern in astrocytes.

Upon increased neuronal activity glutamate is released to the synaptic cleft. A reactive increase in the expression of glutamate uptake transporters within astrocytes allows for the adaptation to the increase in glutamate concentrations at the synapse, which will then feedback on neuronal excitability and LTP induction. In addition, glutamate activates also metabotropic glutamate receptors in astrocytes which regulate glutamate release further. We clarified this paragraph on  page 4, line 164-179.

  1. The contents of conclusions and future directions are ambiguous. The authors seem to emphasize an importance of functional study on astrocyte in juvenile stress model. However, the point is not clear especially from page 10, line 439 to the end.

 To date, most of the studies investigating a possible role of astrocytes after early-life stress describe merely differences in the expression of astrocyte markers such as GFAP or S100ß. However, as described in our review, astrocytes can influence neuronal plasticity on many levels, including neurotransmitter uptake and metabolism, gliotransmission and metabolic coupling. Moreover, upon neuronal activity astrocytes react with changes in gene expression, structure and gliotransmitter release. Such alterations in different functional domains are barely investigated after early-life stress. A few studies have begun to investigate the consequences of juvenile stress on astrocytic neurotransmitter uptake and metabolism and its role in STP and LTP. Similar studies need to be conducted for other models of early life stress (maternal separation, limited nesting and bedding) and future studies should also address other functional domains of astrocytes, such as gliotransmission or structural plasticity, in the different early-life stress models. By that, new therapeutic targets can be identified for the therapy of stress-induced psychopathologies like anxiety disorders and depression.

To strengthen this line of argumentation for suggesting future experiments, we rewrote large parts of the conclusion and future directions section, starting from page 10, line 461, and hope we could explain our suggestions for future research on astrocytes in early-life stress more clearly.

  1. Overall, the neural plasticity of hippocampus in stress animal model is a main content of this manuscript although they highlight astrocyte. I think the authors had better add the "hippocampus" in the title.

 We changed now the term “plasticity” in title and abstract to “hippocampal plasticity” (page 1, line 2 and line 24)

Reviewer 2 Report

This is a nice review of astrocytes and early life stress.  It is topical, thorough, and relevant to the readers of the journal. 

Author Response

We thank the reviewer for her/ his supportive comments.